# Gifsplanation via Latent Shift:
# A Simple Autoencoder Approach to
# Counterfactual Generation for Chest X-rays

**Joseph Paul Cohen**[1,2,3]                    JOSEPH@JOSEPHPCOHEN.COM
**Rupert Brooks**[4]                            RUPERT.BROOKS@NUANCE.COM
**Sovann En**[4]                                SOVANN.EN@NUANCE.COM
**Evan Zucker**[1,2]                            EZUCKER@STANFORD.EDU
**Anuj Pareek**[1,2]                            ANUJPARE@STANFORD.EDU
**Matthew Lungren**[*1,2]                       MLUNGREN@STANFORD.EDU
**Akshay Chaudhari**[*1,2]                      AKSHAYSC@STANFORD.EDU

[1] *Stanford University Center for Artificial Intelligence in Medicine & Imaging*

[2] *Stanford University Department of Radiology*

[3] *Mila, Quebec AI Institute*

[4] *Nuance Communications*

## Abstract

**Motivation:** Traditional image attribution methods struggle to satisfactorily explain predictions of neural networks. Prediction explanation is important, especially in medical imaging, for avoiding the unintended consequences of deploying AI systems when false positive predictions can impact patient care. Thus, there is a pressing need to develop improved models for model explainability and introspection.

**Specific problem:** A new approach is to transform input images to increase or decrease features which cause the prediction. However, current approaches are difficult to implement as they are monolithic or rely on GANs. These hurdles prevent wide adoption.

**Our approach:** Given an arbitrary classifier, we propose a simple autoencoder and gradient update (Latent Shift) that can transform the latent representation of a specific input image to exaggerate or curtail the features used for prediction. We use this method to study chest X-ray classifiers and evaluate their performance. We conduct a reader study with two radiologists assessing 240 chest X-ray predictions to identify which ones are false positives (half are) using traditional attribution maps or our proposed method.

**Results:** We found low overlap with ground truth pathology masks for models with reasonably high accuracy. However, the results from our reader study indicate that these models are generally looking at the correct features. We also found that the Latent Shift explanation allows a user to have more confidence in true positive predictions compared to traditional approaches ($0.15\pm0.95$ in a 5 point scale with $p=0.01$) with only a small increase in false positive predictions ($0.04\pm1.06$ with $p=0.57$).

## 1. Introduction

It is important to understand why a neural network model is making a prediction to ensure that it is using features that we would expect as well as discovering what unknown features a model is using. Typically 2D attribution maps are used which are based on a 1st order approximation of the neural network (Simonyan et al., 2014) but these have limitations as they may just represent edges (Adebayo et al., 2018) or simply not indicate the features that are really being used (Viviano et al., 2020; Arun et al., 2020a,b).

---

[*] Contributed equally

Recently, the idea to visualize predictions via exaggerating features that change the predictions of a model has been discussed by Singla et al. (2020, 2021). This exaggeration is the result of a neural network's ability to hallucinate features (Cohen et al., 2018; Baumgartner et al., 2018) which is known to be controllable (Mirza and Osindero, 2014; Wang et al., 2020). Instead of simply generating images of a specific class, these exaggeration methods can explain the specific features used by a classifier to make each prediction. This is valuable in detecting when a model predicts using incorrect spurious correlates to ensure it is right for the right reasons (Ross et al., 2017; Zech et al., 2018). While most image pathology prediction models have expected causal relationships where specific image regions explicitly lead to the classification label (Enlarged heart $\rightarrow$ Cardiomegaly), models predicting future risk (e.g. 5 year mortality) do not have such a known causal relationship. In these scenarios, we can learn which features are being used with these methods and viewing the counterfactual image.

However, there are two major downsides to existing approaches to this task which limit their adoption. 1) They are based on GANs (Goodfellow et al., 2014) which can be very difficult and time consuming to train because of loss function stability and hyperparameter sensitivity. 2) They are monolithic models that require the generative and discriminative components to be trained together which prevents working with existing pretrained models.

One would prefer an approach which is modular, as simple as possible to implement, and able to work with any existing classifier as a drop in replacement for gradient based attribution maps.

Our approach requires a latent variable model, such as a simple autoencoder $D(E(x))$ where $E$ is the encoder and $D$ is the decoder, and a classifier $f$ which predicts a target $y$ as follows: $y = f(x)$. The latent variable model and the classifier are trained independently without any special considerations except for being differentiable. We specifically use an autoencoder because it is simple to implement and train and we believe this will increase adoption of this method.

Once these models are trained, an explanation can be computed as follows. An input image $x$ is encoded using $E(x)$ producing a latent representation $z$. Perturbations of the latent space are computed for a classifier $f$ in Eq 1 which is then used to produce $\lambda$-shifted samples shown in Eq 2.

$$z_\lambda = z + \lambda \frac{\partial f(D(z))}{\partial z} \qquad (1) \qquad\qquad x'_\lambda = D(z_\lambda) \qquad (2)$$

The image $x'_\lambda$ now is expected to produce a higher prediction such that $f(x'_\lambda) > f(x)$. From here we can generate multiple $x'_\lambda$ images to exaggerate or remove features which result in a prediction (explored in §4.2). These images can be stitched together into short videos (gifs) that help to explain why a prediction was made and what representation the classifier had about the concept. Examples available online[1].

An overview of this method is shown in Figure 1. With this approach it is important to keep in mind that this method is limited by the latent representation of the autoencoder. If the decoder is not expressive enough then it will not be able to correctly represent the features used by the classifier. Fortunately, this approach allows multiple classifiers to be

---

1. https://mlmed.org/gifsplanation/

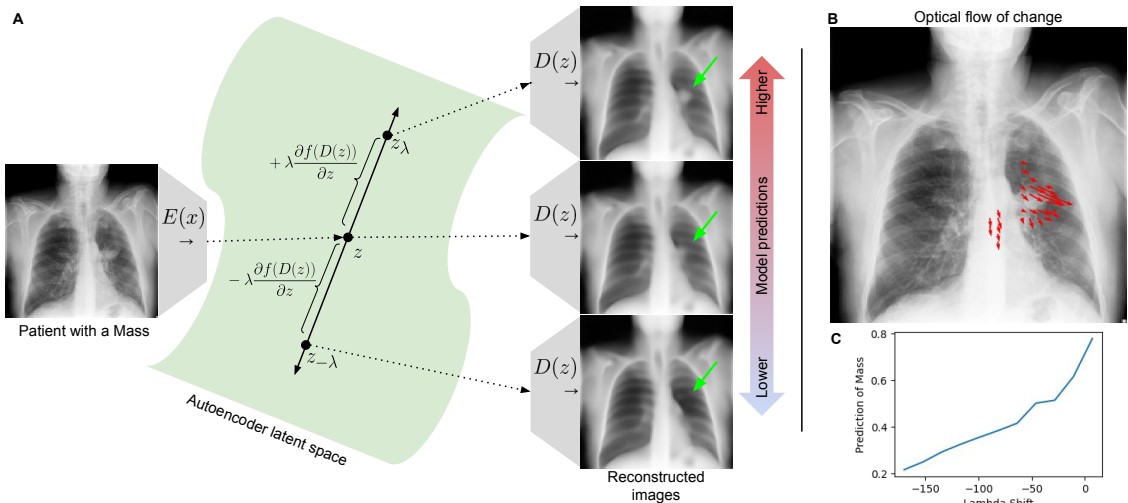

Figure 1: A) Overview of the Latent Shift method. The encoder and decoder parts of the autoencoder are shown in gray. The classifier $f$ predicts if the CXR has a 'Mass'. The image is input from the left and multiple different versions are reconstructed. B) Optical flow computed on a sequence of generated images to visualize what is changing. C) The prediction of 'Mass' changes as the $\lambda$ value changes the reconstructed image.

compared with a fixed autoencoder (or the choice of latent variable model) and allows a clear understanding about the different representations between the models.

In essence we want the exact opposite of an adversarial attack. If we were just modifying the image using the gradient $\frac{\partial f(x)}{\partial x}$, which is a traditional adversarial attack, the modification would be imperceivable and distort the image by selecting spurious pixels which happen to have an impact on the target variable. Our approach regularizes this process using a fixed decoder to keep the image on the data manifold and prevent these spurious pixels from changing. Overall, we seek to modify only the most semantically meaningful pixels that lead to a particular classification output. The contributions of our work follow:

1. Propose a simple and elegant approach to counterfactual generation as well as a way to calculate a replacement for a traditional 2D attribution map.

2. Explore the attribution of chest X-ray predictions using this method compared to traditional methods in terms of IoU overlap with expert masks and cascading randomization analysis.

3. Study how this method impacts a radiologist's ability to interpret the prediction of a model compared to traditional attribution methods when presented with false positive predictions.

## 2. Related Work

The idea of decoupling models was raised before and these approaches are similar in spirit to our approach in how they walk around the latent space although they have different formulations and utilize GANs. Schutte et al. (2020) learned a small function to map the latent variable to a predicted target and use it to transform the latent variable. Joshi et al. (2018) moves in the latent space based on the classifiers loss function in order to change the class of the image. They recursively modify the latent variable until the class changes.

## 3. Protocol and Materials

### 3.1. Chest X-ray classifiers

Three DenseNet121-based classifiers from existing publications were used. There is no requirement for this specific architecture but there are not many publicly available chest X-ray models. Two models are from the paper (Cohen et al., 2020a) referred to as the XRV-all and XRV-mimic_ch. The XRV-all model is jointly trained on 7 CXR datasets (NIH, PC, CheX, MIMIC-CXR, Google, OpenI, RSNA which are described in Appendix §A). The XRV-mimic_ch model is trained on only MIMIC-CXR (Johnson et al., 2019). The other model is from the JF Healthcare group (Ye et al., 2020) which was built for the CheXpert challenge (Irvin et al., 2019) and at one point was ranked 1st on the leaderboard.

### 3.2. Generating 2D attribution maps

There are a few ways to generate a 2D Latent Shift attribution map which would be comparable to a typical attribution map. Here we will discuss the latentshift-max method which was found to work best. This method takes a sequence of $x'_\lambda$ images between a specific $\lambda$ range (discussed in §4.2). First the absolute difference between the non-shifted reconstruction $x'$ and each of the shifted $x'_\lambda$ images is computed. Then the maximum difference at a per pixel level is computed to produce the final attribution map. Intuitively, this captures the maximum change as the result of the shift. More options for this conversion are discussed in Appendix §B.

### 3.3. Baseline attribution methods

The baseline method of *input gradients* (referred to as *grad*) computes the absolute gradient of the input with respect to the prediction made for all images of the positive class $|\frac{\partial \hat{y}_1}{\partial \mathbf{x}}|$ (Simonyan et al., 2014). The method *Guided Backprop* (Springenberg et al., 2015) (referred to as *guided*) tries to ignore gradients that cancel each other out by only backpropagating positive gradients. The method *Integrated Gradients* (Sundararajan et al., 2017) (referred to as *integrated*) works by integrating gradients between the input image $x_i$ and an all-zero baseline image.

### 3.4. Mask annotation datasets and IoU calculation

Expert mask annotations were used to evaluate attribution maps. Bounding boxes from the NIH dataset (Wang et al., 2017) were used for Atelectasis, Cardiomegaly, Effusion, and Mass. Segmentation masks from the RSNA Pneumonia Challenge (Shih et al., 2019) were used for Lung Opacity. Segmentation masks from the SIIM-ACR Pneumothorax Challenge (Filice et al., 2020) were used for Pneumothorax. Additional details in Appendix §A.3.

To fairly compute an IoU value (intersection over union; $\text{IoU}(\text{mask}, \text{img}) = \frac{\text{mask} \cap \text{img}}{\text{mask} \cup \text{img}}$) for the 2D attribution methods we followed (Viviano et al., 2020) where a binarized attribution map is created such that the top $p$ percentile pixels were set to 1, where $p$ is dynamically set to the number of pixels in the ground truth mask that it is being compared to.

## 4. Experiments

All source code[2] and datasets (see §A) are publicly available. The classifiers, autoencoder and their respective pre-trained weights as used in this work are available in TorchXRayVision 0.0.24 (Cohen et al., 2020b). PyTorch 1.6.0 (Paszke et al., 2017) and Captum 0.3.0 (Kokhlikyan et al., 2020) were used for model training and feature attribution, respectively.

---

2. `https://github.com/mlmed/gifsplanation`

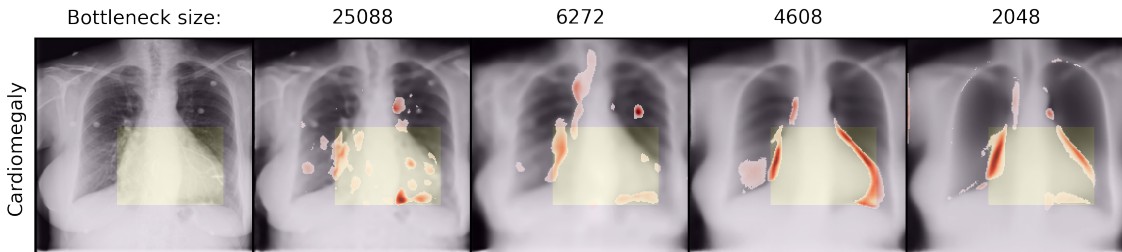

Figure 2: The latentshift-max method to generate 2D attribution maps is applied to the same image using autoencoders which vary in bottleneck size. The top 95% of explanation pixels are shown.

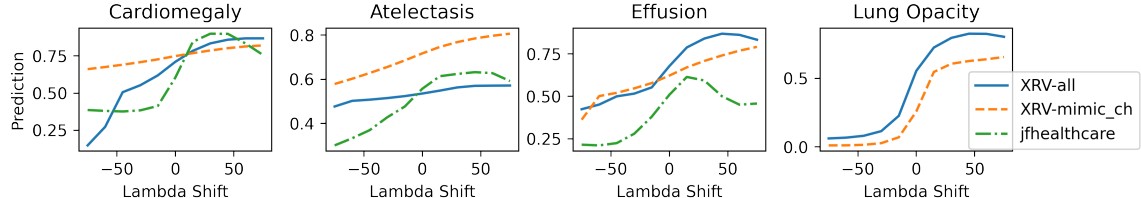

Figure 3: Example of the pathology prediction as $\lambda$ is moved along the latent shift axis for 3 different classification models on the same image. The same autoencoder is used in all cases. At $\lambda=0$ the prediction is the classifiers output on the unmodified reconstructed image.

## 4.1. Autoencoder architecture and training

Keeping with our goal to build the most straightforward model, a ResNet (He et al., 2016) convolutional autoencoder was used as it is able to achieve high fidelity image reconstruction and is relatively easy to implement. An elastic (squared + absolute) loss was used to capture both large and small features. This model was trained on 4 large datasets NIH, PC, RSNA, and MIMIC.

The bottleneck of the autoencoder is a major variable in the quality of the explanations. In Figure 2 the bottleneck size is varied and latentshift-max images are computed using the XRV-all model to predict Cardiomegaly (an enlarged heart). Looking qualitatively at the generated image explanations and their corresponding videos we observe that a large bottleneck results in spotty changes in the region of interest but they don't appear to clearly vary the pathology. At smaller bottleneck sizes the size if the heart appears to be controlled. However, if it is too small then small features, such as the ribs, are lost. In further experiments a ResNet101 with a bottleneck size of 4608 is used.

Unexpectedly we find that larger bottleneck sizes have a higher IoU but they do not result in a better explanation when viewed qualitatively. The shifted images do not appear to have a smooth transition between each other and changes appear unrelated to the pathology. This brings to question how well the IoU analysis captures the quality of these approaches. During training we find that as validation MAE decreases later in training the IoU also goes down. This indicates that the specific reconstruction error seems sufficient only initially in training. Likely towards the end of training, minimizing the small details hurts the ability to control major features of the images. See Appendix §D for more plots.

## 4.2. Determining the $\lambda$ range

When making changes to the latent representation it is important to control the extent of the change. Too little and the difference between the images won't be significant enough to change the prediction of the model. Too large and the image will become too distorted and won't represent the pathology.

In Figure 3 the latent representation is varied by different $\lambda$ values for three different models on four different tasks. Here the direction of the change in the latent space is defined

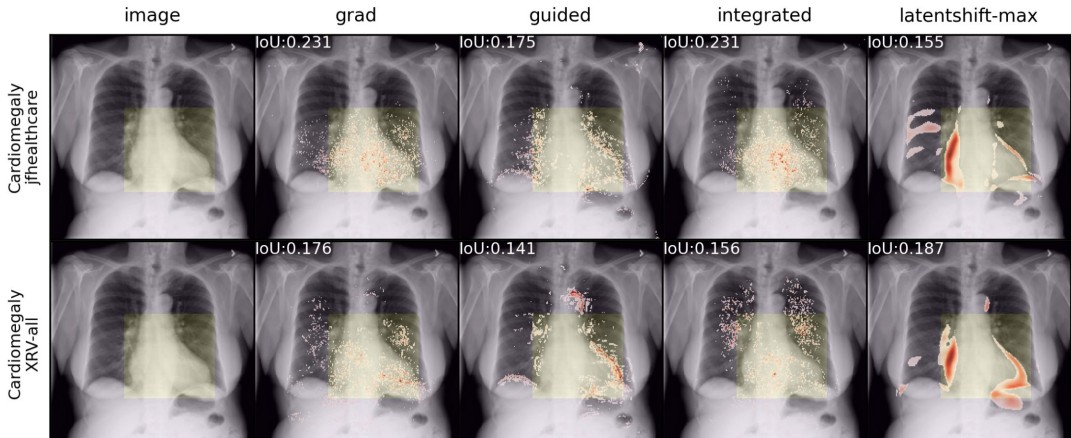

Figure 4: The XRV-all and jfhealthcare models make positive predictions on images for Cardiomegaly. These predictions are explained using multiple 2D attribution maps. A expert bounding box is shown for Cardiomegaly in yellow. No Gaussian blur is applied to these attribution maps.

by the gradient computed for each model. We observe there is variation between how the prediction changes for each model. The smoothness here is a sign that the representation is good. Surprisingly the dynamic range of the predictions between these tasks is similar. We observed that this range is decoder specific and different decoders will have much larger or smaller dynamic ranges. When creating sequences of images we utilize a simple iterative search algorithm to determine the lower and upper $\lambda$ values (see Appendix §C). The lambdas are chosen such that the prediction decreases by 30% and increases by 5%. We find the pathologies seem most clear when the image sequence removes the pathology in contrast to prior work which exaggerates it.

### 4.3. Qualitative 2D attribution map comparison

In Figure 4 qualitative results are shown when varying the model and pathology across multiple attribution methods. One very notable difference is that this method produces a smoother attribution map without blurring. The gradient based approaches have a speckled pattern which is typically alleviated using Gaussian blur. Between the two models evaluated we can see that similar regions are highlighted but they also have distinct differences. This variability is a powerful aspect of this method because we can study the different features used between models. Here it appears that the JF Healthare model mostly looks at the right side (chest right = image left) of the heart while the XRV-all model looks at both sides. This is also confirmed by looking at the generated videos. 2D images only present a small amount of information that this method provides. Videos and images can be seen side by side at this URL[3]

### 4.4. Quantitative IoU comparison

The different 2D attribution maps are compared based on their IoU in Table 1. This experiment confirms that this method produces similar attributions as other methods. While two models achieve reasonable AUC scores for Pneumothorax their IoU scores are extremely low which indicates either the pathology is predicted using spurious features, the bounding boxes are wrong, or that the model is predicting using some confounding pathology. The overall low scores yet high AUC bring into question the validity of using bounding box or mask information to evaluate attribution methods.

---

3. `https://mlmed.org/gifsplanation/`

Table 1: The IoU and AUC is evaluted for 4 attribution methods are studied over 3 models. For each task the IoU was calculated as the mean over 80 samples. The AUC was calculated as the mean over 2048 samples from the same dataset. Note that we compute the best case IoU (see §3.4).

| Task | Dataset | Example | Model → 2D Method | XRV-all | | XRV-mimic_ch | | JF Healthcare | |
|---|---|---|---|---|---|---|---|---|---|
| | | | | AUC | IoU | AUC | IoU | AUC | IoU |
| Atelectasis | NIH | | grad | | 0.07±0.07 | | 0.06±0.07 | | **0.13±0.10** |
| | | | guided | 0.78 | 0.09±0.08 | 0.70 | 0.04±0.04 | 0.77 | 0.10±0.07 |
| | | | integrated | | 0.05±0.05 | | 0.04±0.05 | | 0.10±0.09 |
| | | | latentshift-max | | **0.11±0.12** | | **0.08±0.11** | | 0.09±0.09 |
| Cardiomegaly | NIH | | grad | | **0.35±0.05** | | **0.25±0.09** | | **0.45±0.04** |
| | | | guided | 0.90 | 0.28±0.06 | 0.69 | 0.15±0.06 | 0.90 | 0.31±0.05 |
| | | | integrated | | 0.27±0.08 | | 0.15±0.08 | | 0.36±0.09 |
| | | | latentshift-max | | 0.33±0.07 | | 0.21±0.09 | | 0.35±0.09 |
| Effusion | NIH | | grad | | 0.12±0.09 | | 0.08±0.08 | | **0.18±0.10** |
| | | | guided | 0.87 | 0.15±0.09 | 0.80 | 0.06±0.05 | 0.87 | 0.14±0.07 |
| | | | integrated | | 0.11±0.08 | | 0.05±0.06 | | 0.14±0.09 |
| | | | latentshift-max | | **0.16±0.11** | | **0.11±0.11** | | 0.16±0.10 |
| Mass | NIH | | grad | | 0.16±0.14 | | Model does not predict | | Model does not predict |
| | | | guided | 0.82 | **0.19±0.16** | | | | |
| | | | integrated | | 0.13±0.13 | | | | |
| | | | latentshift-max | | 0.14±0.17 | | | | |
| Lung Opacity | RSNA | | grad | | **0.21±0.11** | | 0.13±0.09 | | Model does not predict |
| | | | guided | 0.84 | 0.21±0.12 | 0.75 | 0.09±0.07 | | |
| | | | integrated | | 0.17±0.10 | | 0.08±0.07 | | |
| | | | latentshift-max | | 0.20±0.13 | | **0.15±0.14** | | |
| Pneumothorax | SIIM-ACR | | grad | | 0.01±0.02 | | 0.01±0.02 | | Model does not predict |
| | | | guided | 0.78 | **0.03±0.05** | 0.67 | 0.02±0.03 | | |
| | | | integrated | | 0.01±0.02 | | 0.01±0.01 | | |
| | | | latentshift-max | | 0.02±0.04 | | **0.03±0.07** | | |

## 4.5. Cascading randomization analysis

Adebayo et al. (2018) showed that even visually convincing attribution maps could be misleading and only weakly dependent on the network parameters. We replicate their proposed cascading randomization evaluation. Starting at the classifier end of the network, layer weights are randomized, and the attribution is reevaluated and the correlation computed between the resulting attribution and the original. Intuitively, one expects that the attribution should rapidly become decorrelated. As shown in Figure 5, the correlation with the final attribution

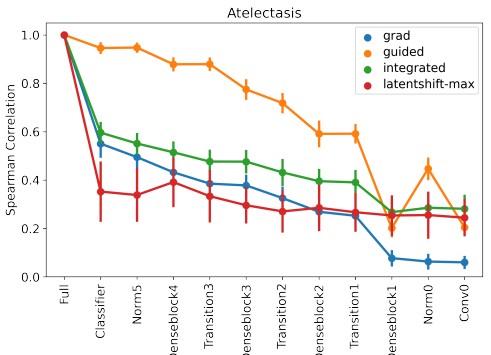

Figure 5: Correlation between attribution generated by different methods when layers in the network are reset.

drops off most rapidly with latentshift-max. Similarly to the findings in Adebayo et al., the guided backprop method produces a very similar attribution even as a significant fraction of the model is reinitialized. The patterns for other pathologies were extremely similar and are shown along with some further details in Appendix H.

## 4.6. Improvement in false positive detection

We performed a reader study to determine if our method can improve the ability to detect false positive predictions (examples in Appendix §J) as well as if the features utilized are correct. For this study we recruited two radiologists (A.J. and E.Z., with 2 and 12 years of experience, respectively). They were presented with 240 images twice, each being predicted as having one of 6 pathologies by the XRV-all model (Atelectasis, Cardiomegaly, Effusion, Lung Opacity, Mass, Pneumothorax). Examples were selected such that 50% were

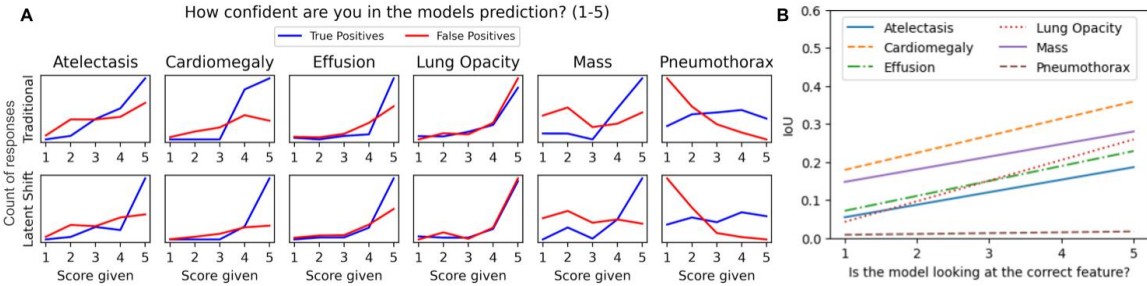

Figure 6: A) Responses to the survey questions split by each pathology. B) Regression lines comparing the IoU for each true positive image with the reader responses.

predicted incorrectly by the model (false positives). An incorrect prediction is defined by having a negative label and a >50% prediction by the model which was calibrated such that a 50% prediction is the operating point of the AUC curve on validation data.

Each sample is viewed using traditional attribution methods (Method A) and the Latent Shift method (Method B). For Method A the radiologist can see all traditional methods at once (Input gradients, Guided Backprop, and Integrated Gradients). For Method B the radiologist can see both the 2D latentshift-max image as well as a gif annimation side by side. Radiologists were asked the following questions on a 5 point Likert scale: "How confident are you in the model's prediction? (1-5)" and "Is the model looking at the correct feature? (1-5)".

The primary study results are shown in Figure 6 and more details can be found in Appendix §F. Overall, for true positive predictions there is a $0.15\pm0.95$ confidence increase using the Latent Shift method ($p=0.01$ using the Wilcoxon signed-rank test). For false positive predictions there is a $0.04\pm1.06$ increase which is not significant ($p=0.57$). We expected false positives to be scored less so these results raise concerns in overconfidence based on model predictions. Although there is the possibility that some of the ground truth labels were wrong.

In the radiologist's feedback (verbatim in Appendix §F.1) they believed that the Latent Shift method was more intuitive and they felt it increased their confidence that the model is looking at the correct feature. They observed that this method looks at the boundaries of the abnormality. One radiologist believed that the model was using the chest tube to predict Pneumothorax instead of looking at the correct area (examples in Appendix §I). This observation is consistent with the IoU analysis and likely because the model input is too low resolution (224x224) to see the small features at the edge of the lung.

## 5. Conclusion

We presented Latent Shift, a simple to implement approach to explain the predictions of models by simulating changes to the input images which increase and decrease the prediction of a classifier. Our approach is designed to be easy to implement in order to increase adoption in other domains and work with existing pre-trained classifiers.

We evaluated Latent Shift and other attribution methods in how well they aligned with ground truth spatial mask information. We found very low IoU values for models with reasonably high AUCs, but with this we cannot conclude which one is in error. The results from our reader study indicate that higher IoU values are correlated with correct features.

We find that the Latent Shift explanation allows a user to have more confidence in true positive predictions compared to traditional approaches. However, we also found that detecting false positive predictions was challenging, which highlights the need for a stronger radiologist-algorithm symbiosis.

## Acknowledgements

We thank Joseph D. Viviano, Chin-Wei Huang, Lan Dao, Jin Long, Pranav Rajpurkar, William J Sehnert, and Levon Vogelsang for useful discussions. This research is based on work partially supported by Carestream Health, the CIFAR AI and COVID-19 Catalyst Grants, and by NIH/NIBIB Grants 75N92020D00018 / 75N92020F00001. Some of the computing for this project was performed on the Sherlock cluster. We would like to thank Stanford University and the Stanford Research Computing Center for providing computational resources and support that contributed to these research results. We thank AcademicTorrents.com for making data available for our research.

## Conflict of interest disclosure

Ashkay Chaudhari has provided consulting services to Skope MR, Inc., Subtle Medical, Chondrometrics GmbH, Image Analysis Group, Edge Analytics, ICM Co., and Culvert Engineering; and is a shareholder of Subtle Medical, LVIS Corporation, and Brain Key; and is on the advisory board for Chondrometrics GmbH and Brain Key; and receives research support from GE Healthcare and Philips not related to this work.

## Ethics

The study conducted in this research has been approved by the ethical review board at Stanford University.

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

## Appendix A. Datasets

### A.1. Autoencoder datasets

NIH, PC, RSNA, and MIMIC

### A.2. Classifier datasets

- XRV-all: NIH, PC, CheX, MIMIC-CXR, Google, OpenI, RSNA

- XRV-mimic_ch: MIMIC-CXR using the CheXpert labeller

- JF Healthcare: CheX

Table A.1: Details of datasets used

| ID | Name | From | Citation | Geographic Region |
|----|------|------|----------|-------------------|
| RSNA | RSNA Pneumonia Challenge | RSNA | Shih et al. (2019) | Northeast USA |
| CheX | CheXpert | Stanford University | Irvin et al. (2019) | Western USA |
| NIH | ChestX-ray8 | National Institutes of Health | Wang et al. (2017) | Northeast USA |
| Google | Google Labelling of NIH data | Google | Majkowska et al. (2019) | Northeast USA |
| MIMIC_CH | MIMIC-CXR with CheX Labels | MIT | Johnson et al. (2019) | Northeast USA |
| PC | PadChest | University of Alicante | Bustos et al. (2019) | Spain |
| OpenI | OpenI | National Library of Medicine | Demner-Fushman et al. (2016) | USA |
| SIIM | SIIM-ACR Pneumothorax Challenge | SIIM-ACR | Filice et al. (2020) | Northeast USA |

Table A.2: Counts of images in each dataset with a positive label for the pathology listed.

| Dataset | Atelectasis | Cardiomegaly | Effusion | Lung Opacity | Mass | Pneumothorax |
|---------|-------------|--------------|----------|--------------|------|--------------|
| NIH | 5728 | 1563 | 6589 | 0 | 3567 | 3407 |
| PC | 3981 | 8420 | 3342 | 0 | 806 | 223 |
| RSNA | 0 | 0 | 0 | 1348 | 0 | 0 |
| SIIM | 0 | 0 | 0 | 0 | 0 | 3576 |
| MIMIC_CH | 10076 | 9831 | 12064 | 13825 | 0 | 2350 |
| CheX | 3195 | 2909 | 8078 | 9736 | 0 | 1802 |
| Google | 0 | 0 | 0 | 221 | 0 | 46 |
| OpenI | 271 | 185 | 120 | 327 | 6 | 14 |

Table A.3: Full listing of counts for bounding boxes and masks available

| Dataset | Task | Mask Type | Count |
|---------|------|-----------|-------|
| NIH | Atelectasis | Bounding Box | 180 |
| NIH | Effusion | Bounding Box | 153 |
| NIH | Cardiomegaly | Bounding Box | 146 |
| NIH | Infiltration | Bounding Box | 123 (not used) |
| NIH | Pneumonia | Bounding Box | 120 (not used) |
| NIH | Pneumothorax | Bounding Box | 98 (not used) |
| NIH | Mass | Bounding Box | 85 |
| NIH | Nodule | Bounding Box | 79 (not used) |
| RSNA | Lung Opacity | Segmentation | 6012 |
| SIIM-ACR | Pneumothorax | Segmentation | 3576 |

## Appendix B. 3D to 2D Construction

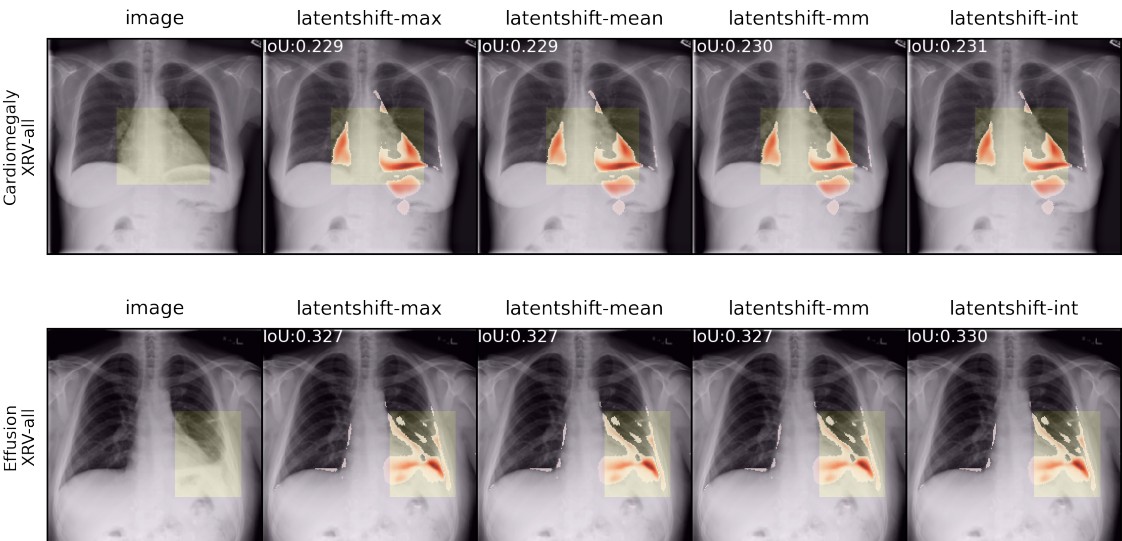

Figure B.1: Examples of the different methods to convert the sequence of images into a 2D. It is hard to find any differences even though they are generated in unique ways.

- **latentshift-mean**: Take the average of all $x_\lambda$ images.

- **latentshift-max**: Take the max distance for each spatial location of all $x_\lambda$ from the image when $\lambda = 0$.

- **latentshift-minmax**: Subtract the lowest $x_\lambda$ from the highest: $|x_{\lambda_{\min}} - x_{\lambda_{\max}}|$.

- **latentshift-sliding interval**: compute the difference between each $\lambda$ step and then average them together.

## Appendix C. Lambda Search

```
lbound = 0
last_pred = classifier(img)
while True:
    img' = compute_shift(img, lbound)
    last_pred = classifier(img')
    if  last_pred < cur_pred
        or initial_pred-0.5 > cur_pred
        or lbound <= -1000
        break
    last_pred = cur_pred
    lbound = lbound - 10
```

## Appendix D. Autoencoder Parameters

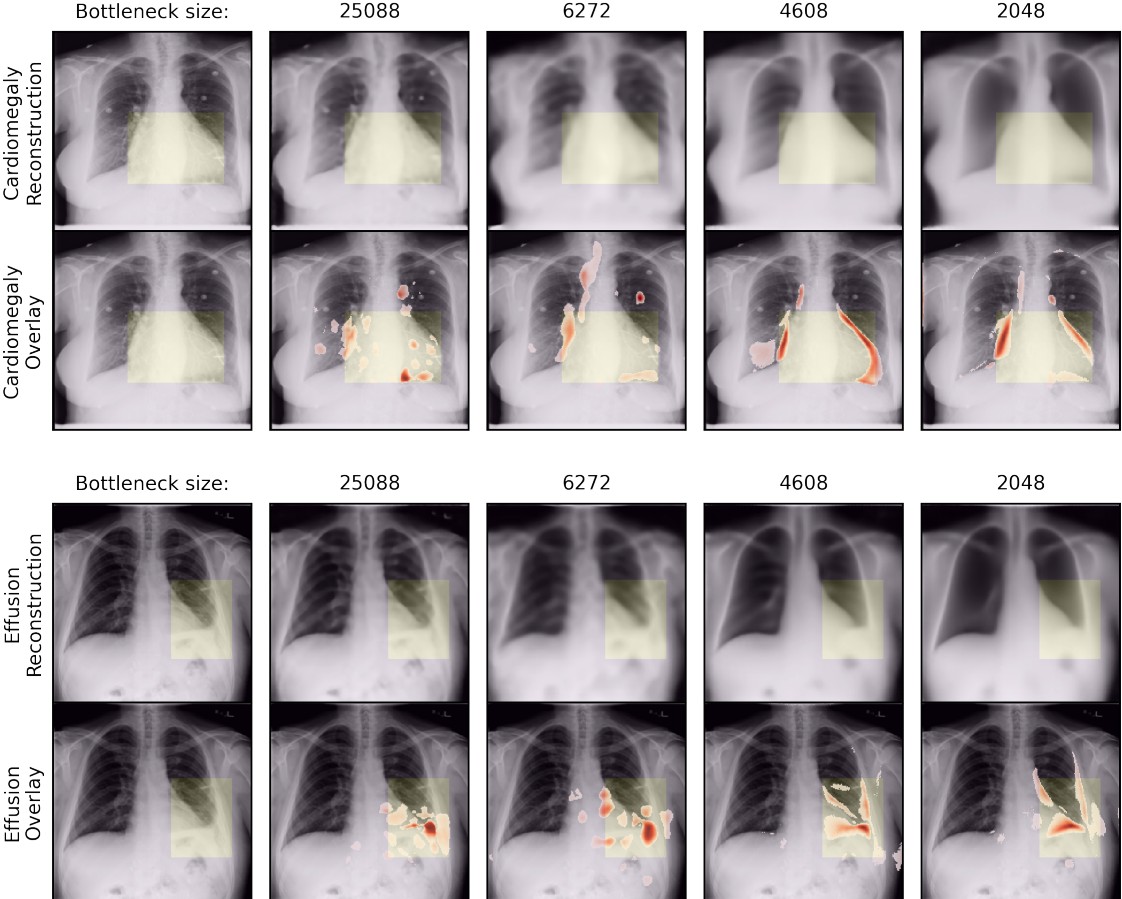

Figure D.1: Varying bottleneck size of the autoencoder. The reconstruction is shown using each autoencoder on the top rows and the latentshift-max method is used to construct a 2D attribution map overlaid on the inout image in the bottom rows.

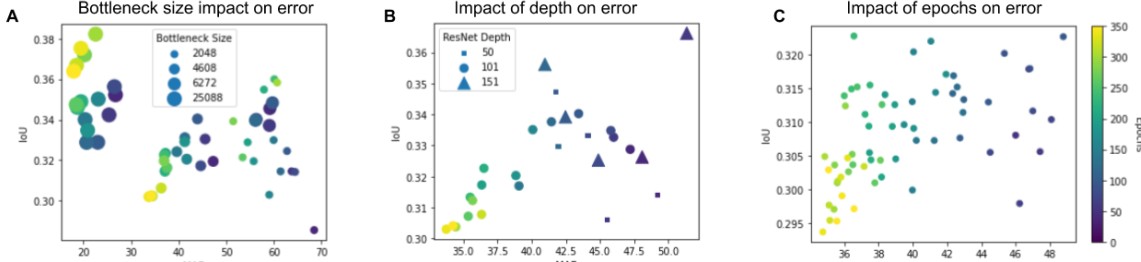

Figure D.2: In A, B, and C each point represents an evaluation of the XRV-all model with a specific autoencoder configuration. The color of each point represents the epoch during training when the evaluation was performed. The model is evaluated on a fixed set of 10 images which contain Cardiomegaly as indicated by NIH bounding boxes. The epoch of training is shown as the color to more fairly compare these networks which converge at different rates. We can see that a larger bottleneck produces a smaller MAE but no strong trend for IoU. In B ResNets of different depths are evaluated and no major trend is found except that potentially a ResNet151 can achieve better IoUs than a ResNet101. However the computational cost is significantly higher and makes this model harder to train.

## Appendix E. Extra IoU Comparisons

There was not space to add this comparison into the main text. We also benchmark the attribution method *Iterative Delete* (Bordes et al., 2018). This approach removes the top first order gradients from the image and reprocesses the image iteratively. This evaluation is performed to serve as a more modern baseline to the baseline attribution methods used in this paper.

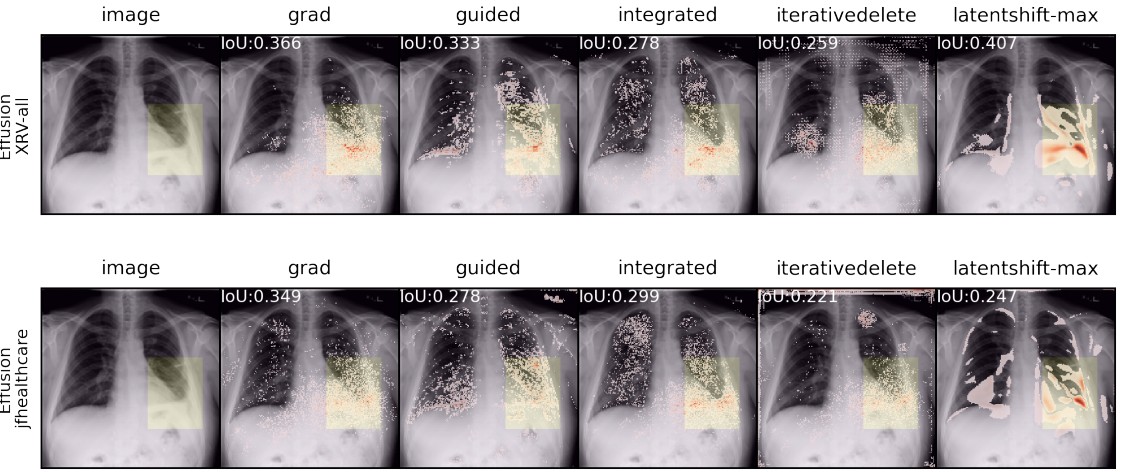

Figure E.1: This is an extension of Figure 4. The XRV-all and jfhealthcare models make positive predictions on images for Cardiomegaly. These predictions are explained using multiple 2D attribution maps. A expert bounding box is shown for Cardiomegaly in yellow. No Gaussian blur is applied to these attribution maps.

Table E.1: IoU evaluation of the Iterative Delete method. Mean is taken over the same 80 samples used in Table 1

| Target | Model Method | XRV-all | XRV-mimic_ch | JF Healthcare |
|---|---|---|---|---|
| Atelectasis | grad | 0.07±0.07 | 0.06±0.07 | 0.13±0.10 |
|  | iterativedelete | 0.05±0.06 | 0.04±0.05 | 0.06±0.07 |
| Cardiomegaly | grad | 0.35±0.05 | 0.25±0.09 | 0.45±0.04 |
|  | iterativedelete | 0.30±0.05 | 0.26±0.09 | 0.30±0.09 |
| Effusion | grad | 0.12±0.09 | 0.08±0.08 | 0.18±0.10 |
|  | iterativedelete | 0.08±0.07 | 0.09±0.09 | 0.10±0.07 |
| Lung Opacity | grad | 0.21±0.11 | 0.13±0.09 | - |
|  | iterativedelete | 0.17±0.09 | 0.13±0.10 | - |
| Mass | grad | 0.16±0.14 | - | - |
|  | iterativedelete | 0.13±0.12 | - | - |
| Pneumothorax | grad | 0.01±0.02 | 0.01±0.02 | - |
|  | iterativedelete | 0.01±0.02 | 0.01±0.03 | - |

## Appendix F. Reader Study

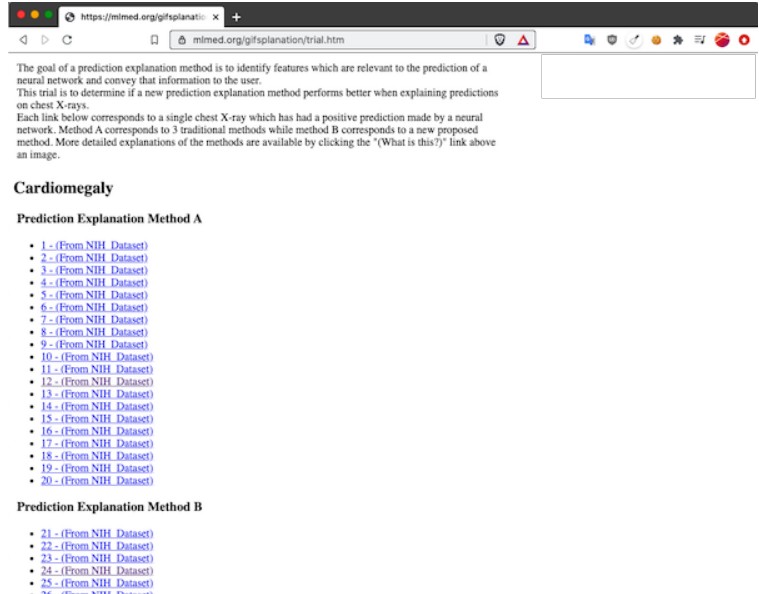

Figure F.1: Screenshots of the primary interface used in the reader study which lists all the images to be studied.

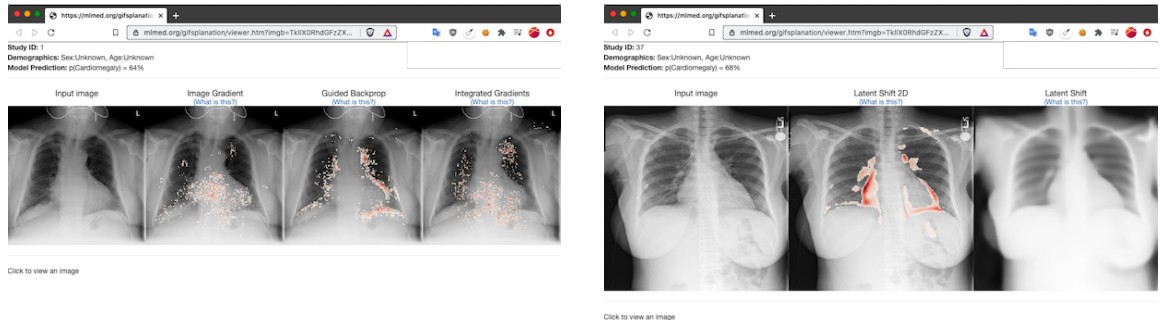

Figure F.2: Screenshots of the per image interface used in the reader study. On the left is the interface with Traditional methods and on the right is when using the Latent Shift method.

### F.1. Reader study feedback

**Reader 1** Some general observations would be that the new prediction method is more intuitive and for most pathologies increases the confidence that the model is looking at the feature a radiologist would look at to make the diagnosis (except for pneumothorax). There were some clear examples where the model made the correct prediction but missed salient findings (e.g., cases 199, 200- predicted mass but did not detect some large masses). Also is interesting that the model in many cases seems to look at the boundaries of an abnormality rather than the actual abnormality or everything else except the abnormality (e.g., contralateral lung) in making predictions so may be a different "interpretation" style."

**Reader 2**

- Latent Shift (B) does much better than gradients (A) approach.

- Within the gradients methods: Image Gradient and Guided Backprop does well, while the highlighted pixels for Integrated Gradients seem to be all over the place (i.e. not good)

- There is a clear correlation between high output prediction probability and better highlighting of important pixels.

- The model is really struggling with pneumothorax - both in terms of prediction and in terms of highlighting correct pixels. This goes for both method A and B. FYI, I did not "count" a resolved pneumothorax as a "positive pneumothorax". I am sure the model sometimes predicts pneumothorax just because there is a chest tube."

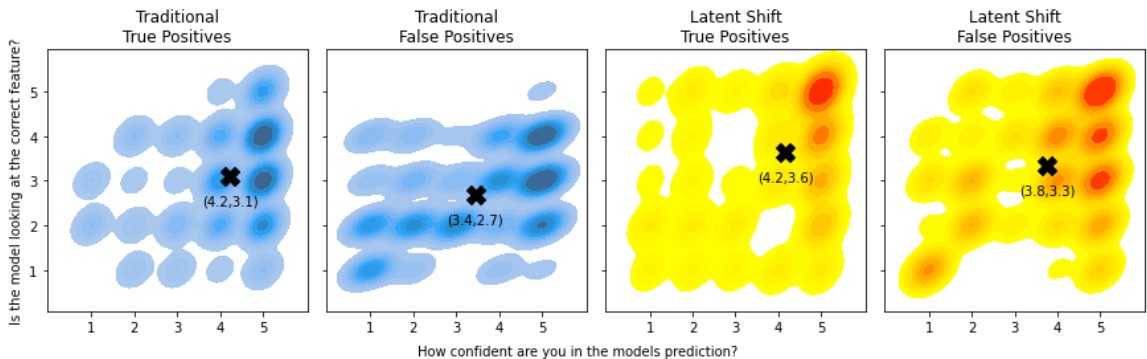

Figure F.3: The relationship between the answers compared between the two methods on both true positive and false positive examples. The **x** mark indicates the mean score.

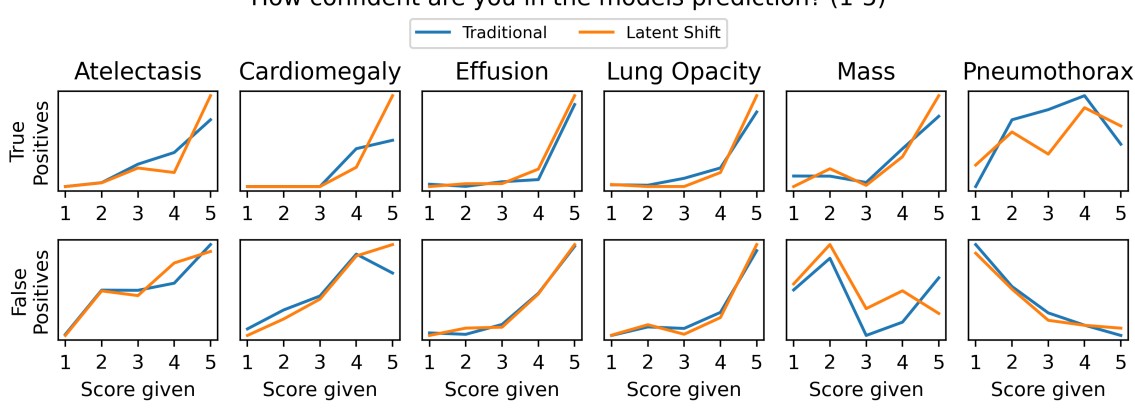

Figure F.4: A different view of the study results showing the counts of survey results.

## Appendix G. Robustness

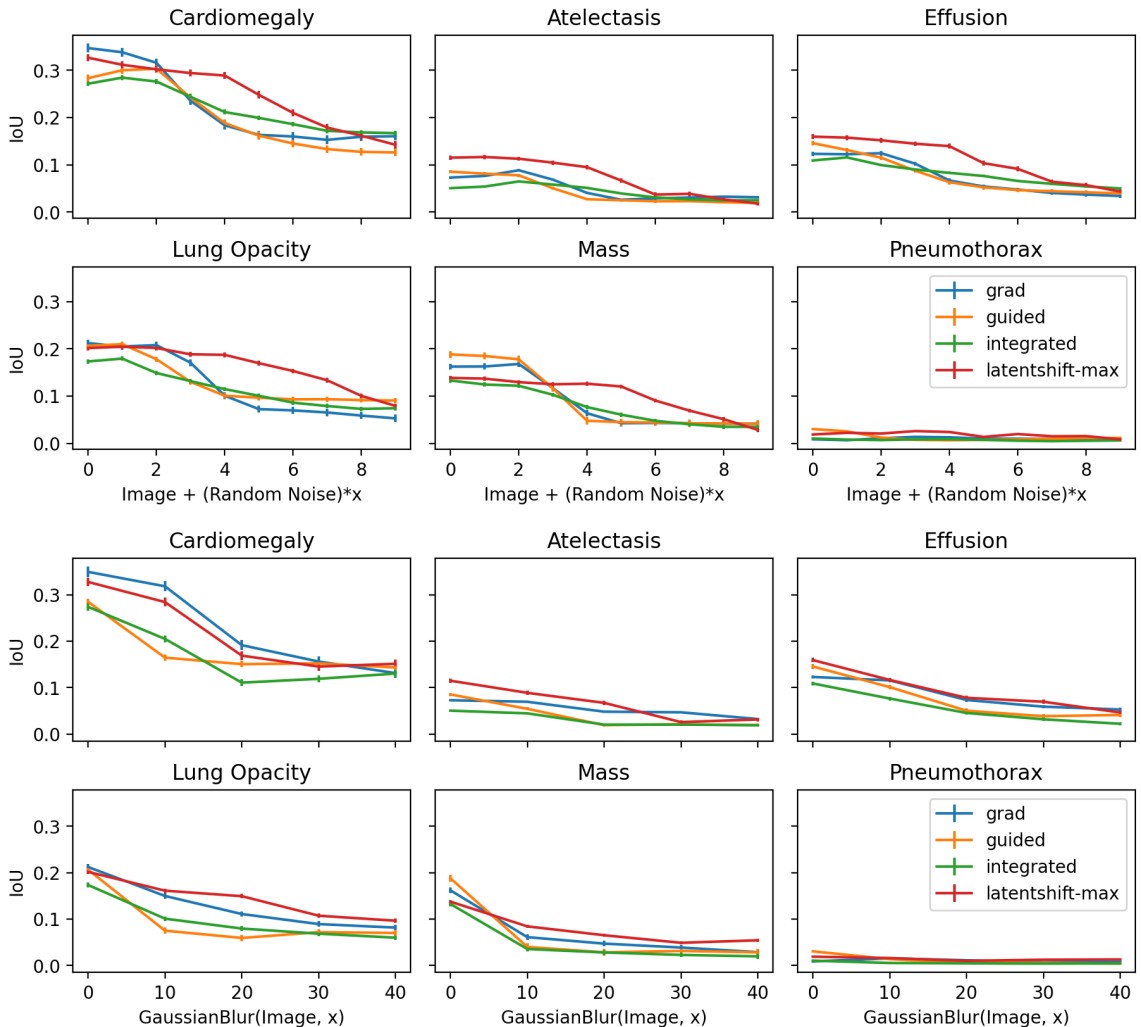

Figure G.1: We study how robust the methods are when adding random noise to the input image at different scales. 80 images are used for evaluation. Images are in the range of [-1024,1024]. Top: Random noise, Bottom: Gaussian Blur.

## Appendix H. Cascading Randomization

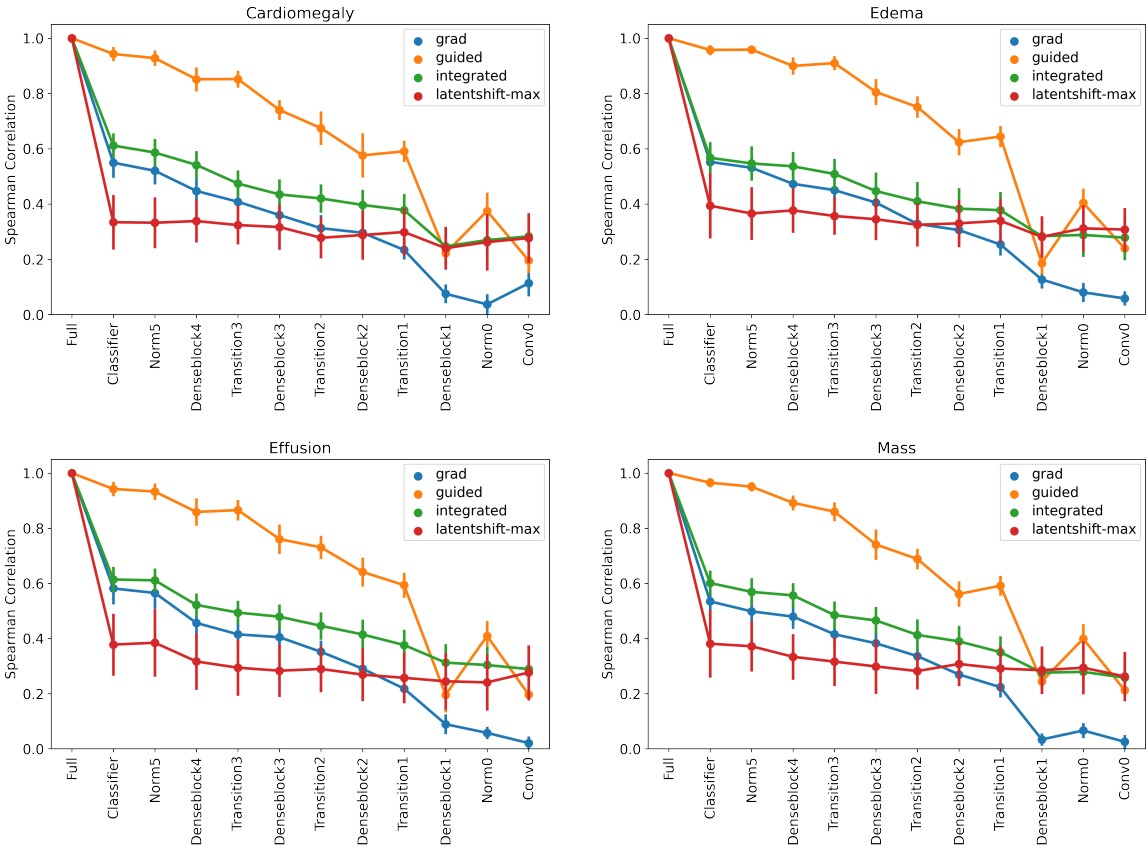

Figure H.1: The cascading randomization sanity check recommended by Adebayo et al. (2018). The test computes Spearman rank correlation between importance of pixels generated by the attribution map as the network is progressively reinitialized. Atelectasis shown in 5, other patterns are very similar. The value is computed over 40 images from the NIH dataset, error bars show standard deviation of the correlation across these images. As the latentshift-max method inherently produces an absolute value map, absolute values are taken of all attribution maps before using this method.

## Appendix I.  Example Explanations

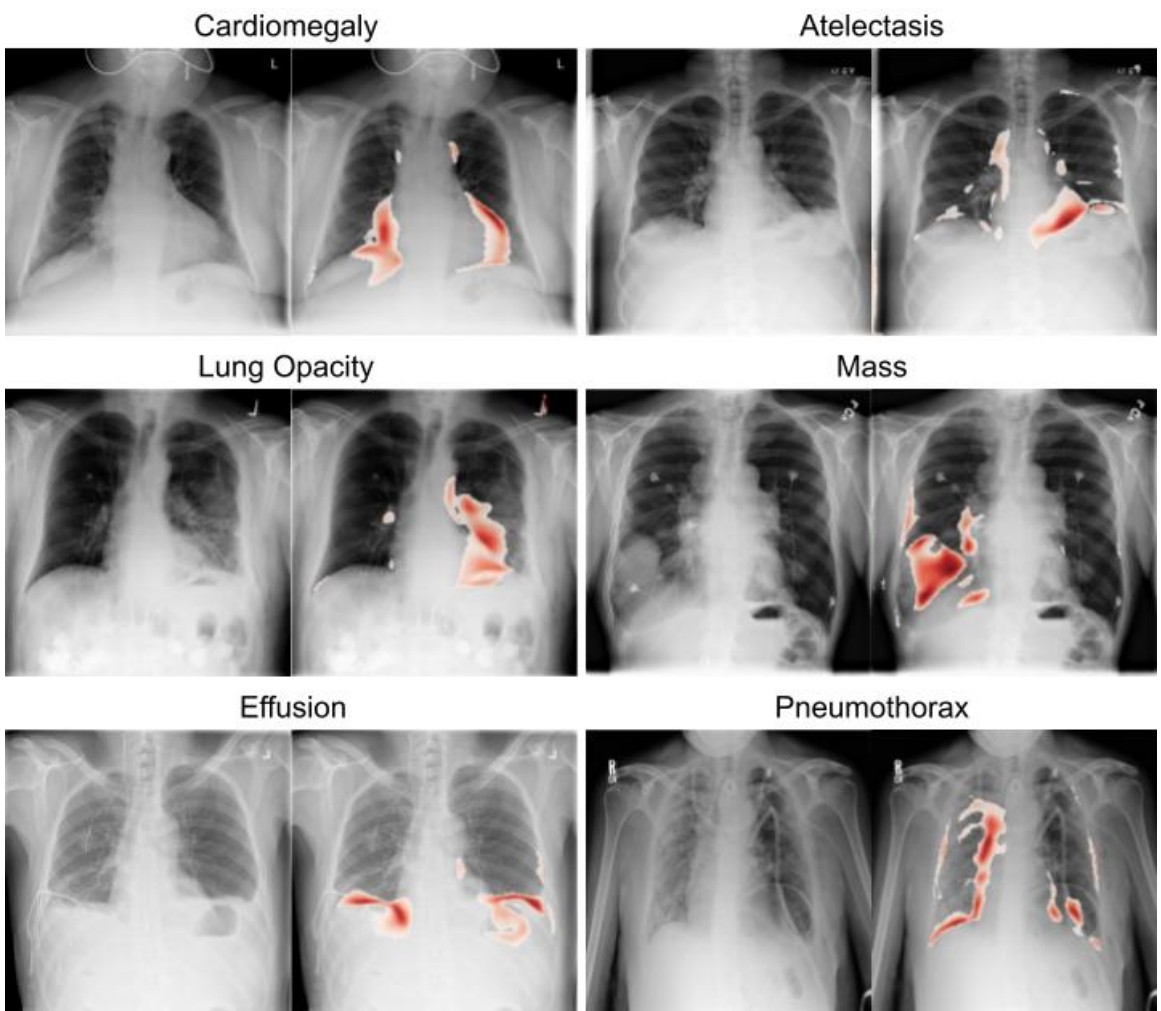

Figure I.1: Extra images of the Latent Shift method applied to different pathologies

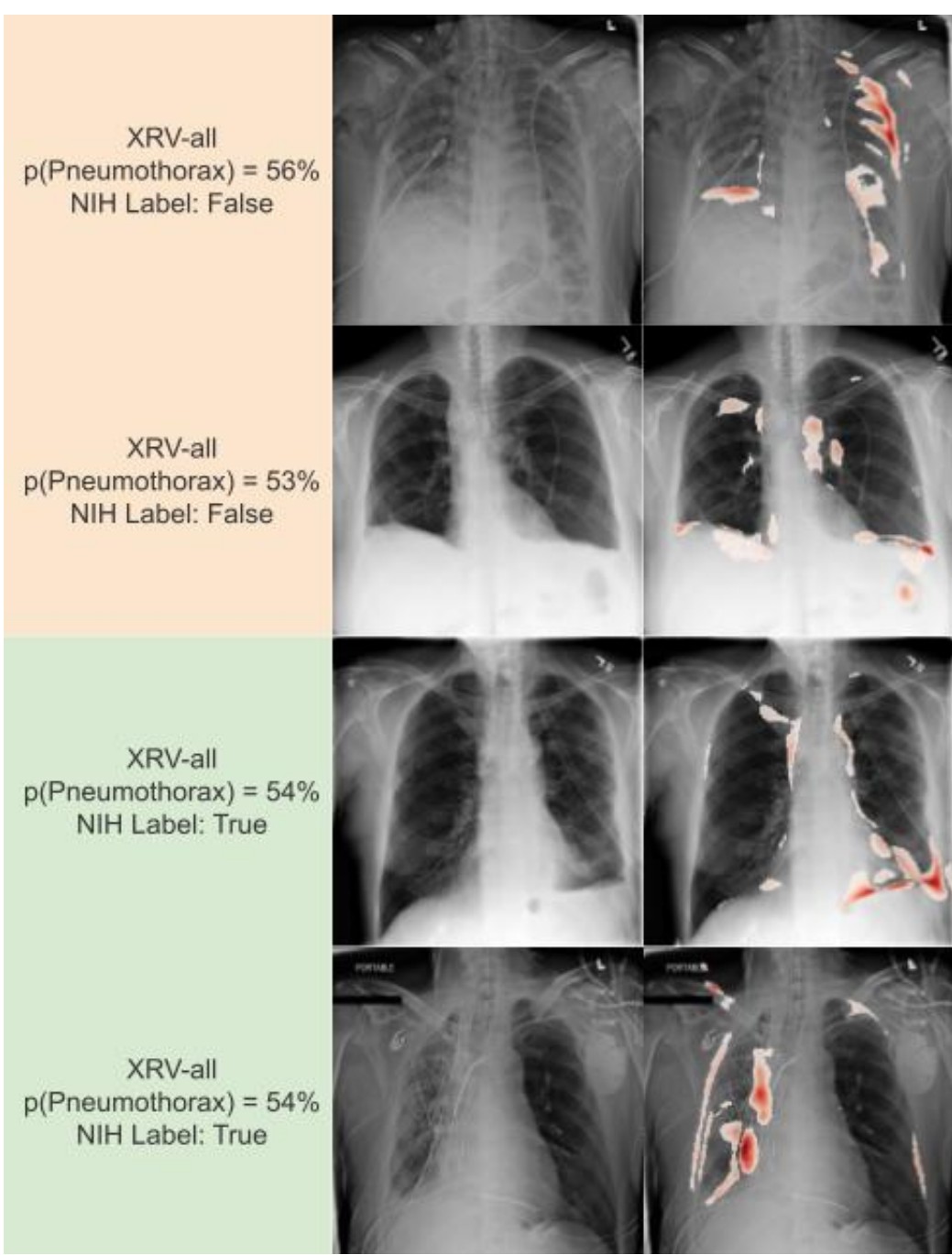

Figure I.2: Example Pneumothorax predictions of both true and false positives.

## Appendix J. False Positive Examples

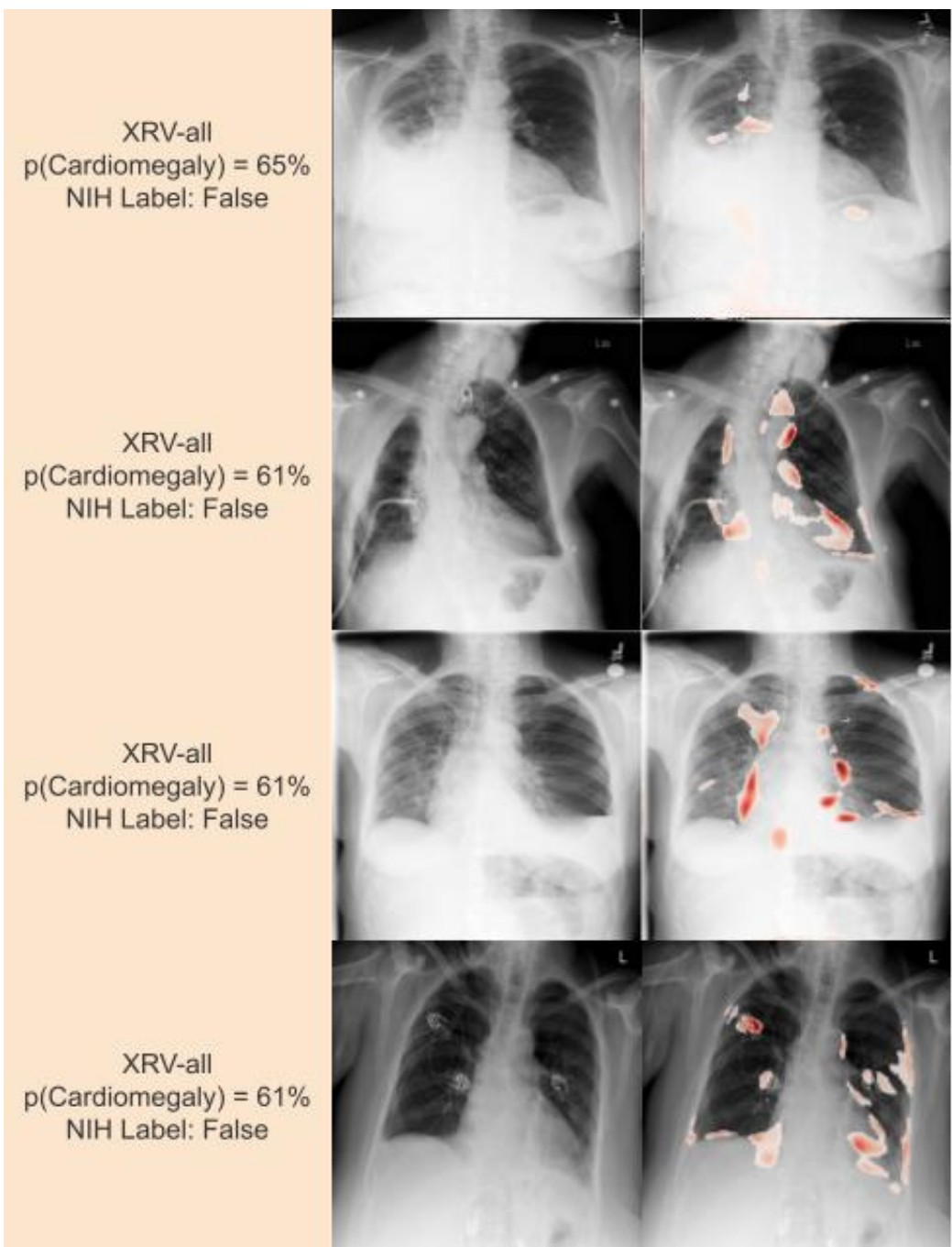

Figure J.1: Examples of false positive examples used in the reader study. The images are from the NIH dataset.

## Appendix K. AE Impact on Classifier Predictions

When the classifier makes a prediction based on the reconstructed image it is often less compared to the input image. The autoencoder reconstruction seems to remove some image features which cause a high prediction.

In order to study the extent of this issue, images for each pathology are evaluated using the classifier and then transformed with the autoencoder (without any $\lambda$ shift) and evaluated using the classifier again. In Table K.1 the results are shown. Each row corresponds to 500 samples. On average predictions were reduced by 0.12 for images with a positive label and reduced by 0.04 for images with a negative label.

Table K.1: Impact of AE transformation on classification prediction.

| Label | Model | Target | Mean $f(x)$ | Mean $f(D(E(x)))$ | diff |
|-------|-------|--------|-------------|-------------------|------|
| Positive | XRV-all | Atelectasis | 0.45 | 0.32 | 0.13 |
| | | Cardiomegaly | 0.53 | 0.40 | 0.13 |
| | | Effusion | 0.55 | 0.45 | 0.09 |
| | | Lung Opacity | 0.64 | 0.53 | 0.11 |
| | | Mass | 0.61 | 0.49 | 0.12 |
| | | Pneumothorax | 0.49 | 0.43 | 0.06 |
| | XRV-mimic_ch | Atelectasis | 0.55 | 0.42 | 0.13 |
| | | Cardiomegaly | 0.60 | 0.57 | 0.03 |
| | | Effusion | 0.60 | 0.48 | 0.12 |
| | | Lung Opacity | 0.59 | 0.39 | 0.20 |
| | | Pneumothorax | 0.54 | 0.45 | 0.09 |
| | jfhealthcare | Atelectasis | 0.47 | 0.36 | 0.11 |
| | | Cardiomegaly | 0.70 | 0.50 | 0.19 |
| | | Effusion | 0.54 | 0.32 | 0.22 |
| Negative | XRV-all | Atelectasis | 0.33 | 0.22 | 0.11 |
| | | Cardiomegaly | 0.19 | 0.13 | 0.07 |
| | | Effusion | 0.22 | 0.19 | 0.03 |
| | | Lung Opacity | 0.29 | 0.28 | 0.01 |
| | | Mass | 0.41 | 0.37 | 0.05 |
| | | Pneumothorax | 0.34 | 0.28 | 0.06 |
| | XRV-mimic_ch | Atelectasis | 0.45 | 0.37 | 0.08 |
| | | Cardiomegaly | 0.42 | 0.40 | 0.02 |
| | | Effusion | 0.33 | 0.26 | 0.07 |
| | | Lung Opacity | 0.38 | 0.24 | 0.14 |
| | | Pneumothorax | 0.46 | 0.40 | 0.06 |
| | jfhealthcare | Atelectasis | 0.30 | 0.34 | -0.04 |
| | | Cardiomegaly | 0.29 | 0.36 | -0.08 |
| | | Effusion | 0.18 | 0.22 | -0.04 |

## Appendix L. Limitations

The autoencoder was developed and trained with the goal of representing specific chest X-ray pathologies. Hyperparameters such as the bottleneck size were specifically tuned to represent the pathologies we studied. We would expect that the resulting autoencoder may not represent other pathologies as well.

The images chosen for the reader study were sampled randomly and may contain multiple different pathologies. Readers are instructed to only consider the specific pathology they are told the model predicted and ignore others.

The models are calibrated such that 0.5 is the operating point of the AUC but often their predictions are lower once they are transformed by the autoencoder. We have performed an analysis to study the extent of this issue in Appendix K. We find on average predictions were reduced by 0.12 for images with a positive label and reduced by 0.04 for images with a negative label.

