# OpenReview forum: "Gifsplanation via Latent Shift: A Simple Autoencoder Approach to Counterfactual Generation for Chest X-rays"
_MIDL.io/2021/Conference — MIDL 2021_

### Official Review · AnonReviewer1 · 2021-03-07

**Confidence:** 5
**Preliminary Rating:** 4
**Recommendation:** Best Paper Award, Oral, Poster

**Summary:**


The authors propose a new method for image attribution: Gifsplanation. An AE is used to transform the latent representation of an input image to exaggerate or curtail the features used for prediction. The method is evaluated on chest X-rays and a reader study with two radiologists assessing 240 chest X-ray predictions to identify which ones are false positives using traditional attribution maps or Gifsplanation. Significance has been statistically explored.

**Strengths:**

This is a fanatic paper about a new methods for deep network introspection.
A reasonably large study with expert observers has been conducted.
The main focus is on confidence identifying false positives. The authors are refreshingly honest about this and show that the increase in false positive predictions is only small (likely because expert observers are heavily biased by reading the images without any additional features).
source code and a nice webpage are available.

**Weaknesses:**

The only weakness is probably the assumption of locally flat embedding manifolds for the latent shift. How far can the shift go before breaking this assumption. A short discussion about this and alternative methods would perhaps help.
Similarities to this article should be discussed: https://res.mdpi.com/d_attachment/applsci/applsci-10-08660/article_deploy/applsci-10-08660.pdf

**Deanonymize Review:**

no

**Detailed Comments:**

It would be nice to discuss (or even test) scalability to other modalities.
A discussion about limitations would also be good.

**Justification Of The Preliminary Rating:**

I opt for strong accept because a) there's nothing really wrong with the paper except the few minor comments above, mainly looking a bit more beyond CXR b) 'weak' scores only introduce more randomness into the reviewing process.

**Paper Type:**

methodological development

**Questions To Address In The Rebuttal:**

probably an outline of the discussions mentioned above

**Special Issue:**

yes

---

> ### Author Response · Authors · 2021-03-17
> **Author Response**
>
> Thanks for your feedback!
>
> > The only weakness is probably the assumption of locally flat embedding manifolds for the latent shift. How far can the shift go before breaking this assumption. A short discussion about this and alternative methods would perhaps help.
>
> We find in the experiment in Figure 3 that it is locally smooth but does fail if we move too far from the initial latent point. This is definitely something that may improve the latent variable model. [Schutte 2020] utilized a StyleGAN and discussed regularizing the latent space to be smooth. It is hard to find space in our paper to add a proper discussion about this. The angle of our paper was to present the most simple approach possible in order to increase adoption of these methods. An extended version of this work should survey how best to regularize the latent variable model. However, it is unclear which metric should be used to evaluate those methods.

---

### Official Review · AnonReviewer3 · 2021-03-08

**Confidence:** 4
**Preliminary Rating:** 3
**Recommendation:** Oral
**Final Rating:** 3

**Summary:**

In this submission the authors propose 1) a novel black box neural network interpretability technique and 2) using this technique suggest displaying salient regions as an animation. To achieve this, an autoencoder is trained separately and then used to modify the query images in the AEs latent space to alter the black box networks prediction. The method is evaluated against baseline saliency techniques in a variety of ways : weakly supervised region detection, the Adebayo et al., 2018 consistency check, and a reader study. The authors find some unwanted results (saliency maps increase confidence in FPs) which are interesting though definitely unwanted and are well discussed. In my opinion, this is a very creative new approach to the long standing problem of obtaining insights into the reasoning of neural networks.

**Strengths:**

Positive points:
 + Interesting, and refreshingly creative approach to saliency maps
 + The authors present an evaluation using a reader study
 + The paper is well written
 + The paper is well motivated

**Weaknesses:**

Negative points:
- The additional benefit of the animations over the simple attribution masks is not evaluated
- The baselines are all from 2017 or older and can be argued to no longer be state-of-the-art
- I have some open questions about the reader study

**Deanonymize Review:**

no

**Detailed Comments:**

1) One of the main contributions is the animation of the saliency map which, as stated previously, I find a very creative approach. However, looking at the videos online I noticed that the animated images are much blurrier than the input images, and I find it hard to make the visual connection between regions in the input image and the animation. I also noticed that the radiologists in the study did not comment on the added advantage of the animations. I believe the paper would benefit from a  evaluation of the additional benefit of the animation over just the overlays, or a discussion thereof.

2) The regular gradient-based saliency techniques compared to in this work, could be argued to not be state-of-the-art  (the most recent baseline approach is from 2017). For instance, in this work (https://openaccess.thecvf.com/content_ICCV_2017/papers/Fong_Interpretable_Explanations_of_ICCV_2017_paper.pdf) it was suggested to modify the input image with multiple gradient steps in the high-dimensional input space (i.e. no autoencoder required), alleviating the problem of spurious pixel attributions to some extent. A comparison against this paper would serve as a stronger baseline and would also individually evaluate the contribution of the autoencoder. Apart from that work there are of course also many other recent works on classifier interpretability for which a comparison would be interesting.

3) Can the curves in Fig. 6A. be interpreted like histograms, i.e. is the y-axis the number of cases in that bin? In Fig 6B, why show regression lines rather than the average IoU for each score? In Fig. 6 in general, wouldn't it be more useful to compare the proposed method versus the baseline saliency methods?

4) From the description in Sec. 4.6 I understood that the 240 samples were split into two groups (A and B) of 120 samples where one was presented with the proposed method and the other with the baseline methods. But then later the authors state that they used the Wilcoxon signed-rank test to assess significance which, I think, requires paired samples. Please clarify this.

5) The confidence increase for TPs and also FPs is an interesting (albeit worrying) finding. Ultimately, it would be very interesting to see if the interpretability enhances the performance of radiologists. The finding that radiologists are more confident in FPs may actually suggest that they perform worse using the proposed saliency map. Did the authors record the radiologists suggested diagnosis? If so, it would be very insightful to see the difference in their performance with the different saliency techniques.

6) It would also be very insightful to see "no saliency maps" as an additional baseline, if it is still possible to obtain this data at this point.

7) In the used dataset, is there always only one pathology in the images? If not, how does this influence the reader study?

8) The modification of the images in the latent space, hinge on the assumption that the latent space is continuous and captures the high-dimensional data in a meaningful way. It is well-known that vanilla autoencoders tend to not have this property as there is nothing enforcing a consistency of the latent space in the training objective (see https://openreview.net/forum?id=HkZN5j-dZH for a reference). I am somewhat surprised that the proposed method works as well as it does. In future work, I suggest to the authors explore using an autoencoder structure which does enforce smooth latent spaces, such as for example a simple variational autoencoder or one of the many newer approaches. This may further improve results and make the model more behaved.

Minor:

9) As a comment for future work, I do not believe the evaluation with IoU of bounding boxes is very meaningful because the objective of classifiers is not to find all of the pixels belonging to a class, but only the fewest pixels necessary for a classification. This is observed empirically and can potentially be explained using information bottlenecks (https://arxiv.org/abs/1703.00810). This effect is demonstrated and discussed in the medical image analysis context in the following paper: https://openaccess.thecvf.com/content_cvpr_2018/html/Baumgartner_Visual_Feature_Attribution_CVPR_2018_paper.html. This means localisation probably shouldn't be attempted using classifier interpretation. I believe this is also confirmed by the overall low IoU scores that were achieved here.

10) In 4.1 the authors state that the autoencoder model is a ResNet. To my understanding, the ResNet is not a autoencoder architecture. Please clarify this.

11) On the bit that states that there is no causal relationship between image regions and future risk of developing a condition. Could the authors clarify what is meant here? Surely, there must be some image region suggesting a risk, otherwise the prediction would be impossible. Or am I missing something here?

12) Regarding "where group A presents traditional attribution methods", I was confused until I saw in the online examples that all of the baselines were presented *simultaneously*. Perhaps, add a word or two emphasising this fact.


**Final Rating Justification:**

Most of my points have been addressed and I am on the brink of giving a "strong accept". The main point holding me back is that the main selling point of the paper, visualising saliency maps with animations, is not evaluated separately. Despite the title, after reading the paper, we do not know if there is in fact any benefit to the animations. Also the radiologists didn't comment on the added benefit of animations. Looking at the online examples personally I find it hard to imagine that these particular animations are more useful than the mask overlays (because not overlaid, and due to bluriness). Together with my other smaller points I stick with my original rating of weak accept.

However, this is an interesting avenue for future saliency research, and the paper has lots of potential for future work. I would definitely want to see this paper at MIDL.


**Justification Of The Preliminary Rating:**

I see this as a proof-of-concept of a new approach to classifier interpretability. The method has some weaknesses, in my opinion, and has potential for improvement and future work. Despite some short-comings, pending the rebuttal, I think this paper will be interesting for the community.

**Paper Type:**

methodological development

**Questions To Address In The Rebuttal:**

Questions are stated in the detailed comments.

In case there is not enough space, please make sure to answer my points about the baselines and the open questions regarding the reader study.

**Special Issue:**

no

---

> ### Author Response · Authors · 2021-03-17
> **Author Response Part 2**
>
> > In the used dataset, is there always only one pathology in the images? If not, how does this influence the reader study?
>
> The images are selected randomly and may contain other pathologies. When the images are presented to the reader it states which pathology is being predicted and they are instructed to focus on this pathology only. This text was added to Appendix section L (Limitations).
>
>
> > As a comment for future work, I do not believe the evaluation with IoU of bounding boxes is very meaningful because the objective of classifiers is not to find all of the pixels belonging to a class, but only the fewest pixels necessary for a classification...
>
> Thanks for these references! The IoU results raise many questions around what we are trying to achieve and how to best address the clinical tasks. We agree that bounding boxes are not perfect but since the goal of the couterfactual generation is to create local changes, the BB were the only way to get some local measure of importance. Yes, rads could have done that but that would have limited how many cases they could read in the context of this study.
> Please share more references if you are able to!
>
> > In 4.1 the authors state that the autoencoder model is a ResNet. To my understanding, the ResNet is not a autoencoder architecture. Please clarify this.
>
> We mean that we used the ResNet idea to build a convolutional autoencoder.  Specifically, it has residual connections around the convolutional blocks. Besides these residual connections it is a traditional autoencoder. You can view the code used for our model here: https://github.com/mlmed/torchxrayvision/blob/master/torchxrayvision/autoencoders.py#L108 (also you can load the exact weights we used with that library)
>
> > On the bit that states that there is no causal relationship between image regions and future risk of developing a condition. Could the authors clarify what is meant here? Surely, there must be some image region suggesting a risk, otherwise the prediction would be impossible. Or am I missing something here?
>
> I think the way it was worded was not clear. Our point is that for say, cardiomegaly, we have a clinical image prior and know that the model should be looking at an enlarged heart  (we know what should cause the prediction). But for mortality predictions for example, we may not know what the cause is and can therefore learn the cause by looking at the explanation (aka counterfactual) to learn what it is. We modified the text to make this more clear.
>
> > Regarding "where group A presents traditional attribution methods", I was confused until I saw in the online examples that all of the baselines were presented simultaneously.
>
> Sorry about that. The text was written too concisely for space. We have expanded that section to make it more clear.

---

> ### Author Response · Authors · 2021-03-17
> **Author Response Part 1**
>
> Thanks for your feedback! We hope we have addressed your questions and concerns. Your comments have helped us to improve the paper. Also, thank you for the many references.
>
> > I believe the paper would benefit from a evaluation of the additional benefit of the animation over just the overlays, or a discussion thereof.
>
> The radiologists were shown both the animation and the overlay. You can view the exact webpage that was presented to the physicians here: https://mlmed.org/gifsplanation/trial1.htm Method B shows the overlay and animation of the latent shift approach.
>
> > The regular gradient-based saliency techniques compared to in this work, could be argued to not be state-of-the-art (the most recent baseline approach is from 2017).
>
> When selecting the baseline methods we went with the most commonly used methods. To address this concern we added appendix E which evaluates a method similar to the one you mention that was published in MIDL2018 (https://openreview.net/forum?id=rJz89iiiM). However, we find that it does not perform as well as the baseline grad method.
>
> > Can the curves in Fig. 6A. be interpreted like histograms, i.e. is the y-axis the number of cases in that bin?
>
> Yes. The figure has been updated to make this clear.
>
> >In Fig 6B, why show regression lines rather than the average IoU for each score?
>
> When just showing the mean, the plot doesn't accurately reflect the trend because of different support of the mean calculation in each of the response levels. The regression smooths over this noise and makes the underlying trend more clear.
>
> > In Fig. 6 in general, wouldn't it be more useful to compare the proposed method versus the baseline saliency methods?
>
> This view didn't seem as informative because we felt the relative difference between true and false positive predictions was a more important observation. Due to limited space we didn't include it in the main text. We added this view to the appendix as Figure F.4.
>
> > From the description in Sec. 4.6 I understood that the 240 samples .. But then later the authors state that they used the Wilcoxon signed-rank test to assess significance which, I think, requires paired samples. Please clarify this.
>
> The 240 examples were looked at twice. Once using traditional methods and once using the latent shift, thereby creating the paired samples required for the statistical test. This has been made clear in the text.
>
> > The confidence increase for TPs and also FPs is an interesting (albeit worrying) finding. Ultimately, it would be very interesting to see if the interpretability enhances the performance of radiologists. The finding that radiologists are more confident in FPs may actually suggest that they perform worse using the proposed saliency map. Did the authors record the radiologists suggested diagnosis? If so, it would be very insightful to see the difference in their performance with the different saliency techniques.
>
> Yes, this would be very interesting and it will be the focus for follow up work! Only the answers to those two questions were collected. All the data collected is here: https://github.com/mlmed/gifsplanation/tree/main/reader-results
>
> > It would also be very insightful to see "no saliency maps" as an additional baseline, if it is still possible to obtain this data at this point.
>
> This would be a great data point to have. We will try as a follow up but we cannot have the data for this paper.

---

> > ### Comment · AnonReviewer3 · 2021-03-18
> > **Clarification**
> >
> > Just a small comment to clarify one of my points.
> >
> >     I believe the paper would benefit from a evaluation of the additional benefit of the animation over just the overlays, or a discussion thereof.
> >
> > > The radiologists were shown both the animation and the overlay. You can view the exact webpage that was presented to the physicians here: https://mlmed.org/gifsplanation/trial1.htm Method B shows the overlay and animation of the latent shift approach.
> >
> > I think the authors misunderstood. The proposed saliency method can be displayed in two ways 1) overlay, 2) animation. The animation is one of the main novelties of this paper. However, from the experiments it cannot be concluded if the animation yields any additional benefit over displaying the mask only, since you show both 1) and 2) to the doctors. Thus, the additional benefit of one of the major novelties is not evaluated.

---

> > > ### Author Response · Authors · 2021-03-18
> > > **.**
> > >
> > > > it cannot be concluded if the animation yields any additional benefit over displaying the mask only, since you show both 1) and 2) to the doctors. Thus, the additional benefit of one of the major novelties is not evaluated.
> > >
> > > Yes, this is a limitation of the study. However, showing both the overlay and the animation together didn't have a very significant improvement in the false positive detection task which would likely be the upper bound of them individually. Also, my intuition is that the overlay provided context when reading the animation. Due to the use of highly trained and busy humans it is difficult to perform many variations of experiments so we grouped them together. We are not able to perform this evaluation before the end of the rebuttal period. In the paper we word our claims as "the Latent Shift explanation" and not specifically the animation. We can make these claims more explicit (once we are able to edit the paper again) to avoid any misunderstanding. Such as "We also found that the Latent Shift explanation (overlay and animation together) allows a user to have more confidence..."
> > >
> > > With the results from this work our next steps will be to improve the method and quantify the impact of animations vs overlays vs (animations and overlays) vs no explanation. Thank you for your inside to help us prepare future experiments!

---

### Official Review · AnonReviewer2 · 2021-03-09

**Confidence:** 5
**Preliminary Rating:** 3
**Recommendation:** Poster

**Summary:**

The paper proposes to explain the decision of an image classifier, by creating multiple images as an explanation. It does so by gradually transforming the latent representation of an image to change the features that play an important role in the prediction decision. Through experiments, the paper demonstrates the applicability of such an explanation to increase end-user confidence in trusting true positive predictions.


**Strengths:**

1. The proposed approach, allows an end-user to use a pre-trained autoencoder and a given classifier to create multiple perturbations of an input image such that features that are important for the classification model are exaggerated or curtailed.

2. Such an explanation, not only highlights where in the image the classifier is paying attention to make its decision but also highlights how the appearance of those regions should be modified for the classifier to change its decision.

3. The authors performed a human evaluation of their proposed explanations to show their usability.


**Weaknesses:**

In Figure 2, the attribution maps are overlaid on the input image. From the figure, it's not clear how the decoded images differ visually with the size of the bottleneck layer. Also, selecting bottle-neck dimensions that best matched the intended explanation i.e. overlaps maximum with the heart region may result in imposing an inductive biased on the explanation generation process, to produce an explanation that the end-user is more likely to accept.

In Figure 3, it is not clear if the authors consider a single image or a population to generate the trend. To provide quantitative results, authors should consider a population and while considering population, the authors should also plot the error bars to show the variance across different values of lambda. Also, it is not clear what is f(x) for the input images used to derive this plot. At lambda = 0, when there is no perturbation, what is f(x)? Is the input image positively detected for a pathology? For Effusion and Atelectasis, at lambda = 0, f(x) < 0.5.

In Figure 4, the authors qualitatively demonstrate the difference between the two classification models. The example shown in the figure can be cherry-picked. The authors should consider performing extended experiments, such as dividing the ground truth segmentation into left and right lung and then computing left-IoU and right-IoU.

In Table 1, the large variance in IoU measurements demonstrated that there is no clear winner in terms of IoU and all the methods are almost similar in their performance. The authors should consider extended experiments to demonstrate that the highlighted regions are actually being used by the classifier to make its decision. For example, consider modifying the input image with different percentages of masking of the important regions and computing drop in prediction performance.

In Figure 6, what does the y-axis denotes?

The authors mention, that radiologists associate the model's prediction for Pneumothorax with chest tubes. An example demonstrating these findings is missing.




**Deanonymize Review:**

no

**Justification Of The Preliminary Rating:**

The paper is well written with a good method and experiment section. The authors demonstrate different strategies to evaluate their model and demonstrate their strength in explaining the decision of a classifier.


**Paper Type:**

both

**Questions To Address In The Rebuttal:**

The authors should consider providing quantitative results to demonstrate the application of their approach.

The authors should provide details on the prospective application scenario. Does this approach aim to provide a local explanation for a given image and classification decision, or a global explanation explaining the decision for a class label?

**Special Issue:**

no

---

> ### Author Response · Authors · 2021-03-17
> **Author Response**
>
> Thanks for your feedback! We hope we have addressed your questions and concerns.Your questions have helped us to improve the presentation of the work!
>
> > In Figure 2, the attribution maps are overlaid on the input image. From the figure, it's not clear how the decoded images differ visually with the size of the bottleneck layer.
>
> The original figure only plotted over the input image. That figure is now updated to have the reconstructed images shown and more detailed figures are added to Appendix D which show the reconstructed images and the overlays separately.
>
> > Also, selecting bottle-neck dimensions that best matched the intended explanation
>
> We agree. We have added this discussion to the Appendix section L (Limitations).
>
> > In Figure 3, it is not clear if the authors consider a single image or a population to generate the trend
>
> We consider a single input image and have added text to make this more clear.
>
> > At lambda = 0, when there is no perturbation, what is f(x)? Is the input image positively detected for a pathology? For Effusion and Atelectasis, at lambda = 0, f(x) < 0.5.
>
> At lambda=0 what is shown is f(D(E(x))) without any shift. It is the prediction of the reconstructed image without any perturbation. The models are calibrated such that 0.5 is the operating point of the AUC but often their predictions are lower once they are transformed by the autoencoder. We have performed an analysis to study the extent of this issue in Appendix K. We find on average predictions were reduced by 0.12 for images with a positive label and reduced by 0.04 for images with a negative label. We do not feel that this invalidates our results presented but it is a potential limitation of the method. This text has been added to the limitations section in Appendix L.
>
> > In Figure 4, the authors qualitatively demonstrate the difference between the two classification models. The example shown in the figure can be cherry-picked. The authors should consider performing extended experiments, such as dividing the ground truth segmentation into left and right lung and then computing left-IoU and right-IoU.
>
> In Table 1 we present quantitative results comparing three classification models over 6 tasks and 80 randomly chosen samples. In order to strengthen this evaluation we also performed the reader study which uses 240 randomly selected images.
> It would be interesting to perform a more granular study using more detailed mask information but this mask information is not easily available.
>
> > In Table 1, the large variance in IoU measurements demonstrated that there is no clear winner in terms of IoU and all the methods are almost similar in their performance.
>
> For these results we only claim that our method produces "similar attributions as other methods" and do not claim they are better. Because they seem visually better (in light of the reader study results and feedback) it is unclear whether the IoU metric is perfect for such tasks.
>
> > The authors should consider extended experiments to demonstrate that the highlighted regions are actually being used by the classifier to make its decision. For example, consider modifying the input image with different percentages of masking of the important regions and computing drop in prediction performance.
>
> This is an interesting idea to validate that classifiers are predicting using specific areas.
> It is not clear that this approach will help us to determine if the attribution maps reflect the correct pathology because this doesn't allow us to escape using the expert masks. If the network is using spuriously correlated features we would need expert masks to discover this.
>
> > In Figure 6, what does the y-axis denotes?
>
> It is the count of responses. The figure has been updated to make this clear.
>
> > The authors mention, that radiologists associate the model's prediction for Pneumothorax with chest tubes. An example demonstrating these findings is missing.
>
> Examples have been added to the Appendix section I.
>
> > The authors should provide details on the prospective application scenario. Does this approach aim to provide a local explanation for a given image and classification decision, or a global explanation explaining the decision for a class label?
>
> The goal is for a local explanation for a given image and classification decision. We have tried to make the text more clear. Please let us know the best place to make it more clear.

---

### Official Review · ~Bram_van_Ginneken1 · 2021-03-10

**Confidence:** 4
**Preliminary Rating:** 4
**Recommendation:** Oral

**Summary:**

This paper proposes an alternative for the ubiquitous but often unhelpful saliency map: a short movie. The method assumes we use some network with an internal representation of the input image, typically one would use the features in some fully connected deep layer of the network stacked into a vector, and a prediction for the presence of certain abnormalities. A decoder is trained to predict how the image would change to increase or decrease the likelihood of the presence of this abnormality. This is then applied to the classification of chest radiographs for the presence of various abnormalities.

**Strengths:**

I find the basic idea of the paper very appealing. Saliency maps are everywhere in articles on deep learning classification in medical imaging and they are hardly ever useful. Moreover, there are so many saliency map techniques to choose from and they often show completely different saliency maps, which is rather concerning. A movie that exaggerates the abnormal pattern that the network believes is responsible for its output, or removes that pattern, may provide valuable insights.

The paper is also well written, the appendices provide useful additional information, the experiments use public data and the code is shared.

**Weaknesses:**

The results are disappointing. I do not find the movies useful. This may have been caused by the wrong choice of application. See detailed comments. A major limitation that is unrelated to the choice of application is that it seems the method cannot generate high-resolution images. Another more fundamental limitation is that the problem of generating an exaggerated abnormal image is probably ill-posed in most applications: there will usually be many possible ways one can change an image such that it is more likely to be positive for the presence of some abnormality. Consider the example where you'd want to say if a chest x-ray contains a nodule. The gif movie could simply generate nodules anywhere in the image, and all these exaggerations are valid solutions to the problem. This is also a fundamental problem with saliency maps. Imagine a case with five nodules. The saliency map could light up at any single one of those, and it would be a "valid" saliency map.  It could be worth discussing such issues a bit in the discussion.

**Deanonymize Review:**

yes

**Detailed Comments:**

Building decoders that generate realistic high-resolution images is a difficult task. The method proposed here seems not capable of doing that. The three reconstructed images in Figure 1 are very smooth and blurred. In this example there is a very obvious abnormality in the right hilum, so there is no need for any explanation here, and in fact, the lower image where the abnormality should have been removed is not realistic, you never would see such a dark blob next to the hilum, and the hilum itself also looks unrealistic. The only thing this illustration conveys is that the decoder produces rather unrealistic images…

For anything in CXR beyond obvious abnormalities (and we do not need deep learning to detect obvious abnormalities) higher resolution is needed.

Why did the authors choose to work with the labels atelectasis, cardiomegaly, effusion, and mass? Masses are not a good choice because they are usually obvious. Atelectasis is a poor choice because it is usually not a clinically relevant finding. Effusion could be useful, if subtle. In most cases it is not hard to see. Cardiomegaly is an interesting one to discuss a bit further. It namely demonstrates a fundamental problem with the 'gifsplanation' and with the approach that is used in >95% of the (far too large number of) papers written about abnormality detection in CXR: as the authors do in this study, they throw the complete image in a CNN and predict labels. Every radiologist has learned how to detect cardiomegaly: by computing the cardiothoracic ratio and applying the definition of cardiomegaly: a CTR>0.5. To measure CTR you need to draw a horizontal line at the widest dimension of the thorax and the heart, and take the ratio of the lengths. We wrote a paper (https://doi.org/10.1109/ACCESS.2020.2995567) where we demonstrated that the inherent interobserver variability you get when humans have to draw lines means that you cannot get above AUC 0.98. Take one radiologist as the reference and the other as the test, AUC is around 0.98. We built a u-net segmentation of the heart and lungs, explicitly computed CTR from that and that gave AUC 0.98, trained with 800 images. Training a CNN with 62k images gave AUC 0.94, and with 600 images 0.83. The authors in this work do something in between and get an AUC of 0.90 (or even AUC of 0.69 in one of the settings in Table 1). A system to detect cardiomegaly with an AUC of 0.90 is useless. Now an explanation movie wanting to remove cardiomegaly can do two things: make the heart more slender or the thorax dimensions wider. What is the point of having a movie doing one of the two (the former, on the Github page)? In my opinion, we do not learn anything of practical value. As a proof of principle of your method, it is fine.

So my assessment of this paper from an application point of view (CXR analysis) is that the study does not learn us anything new. The presented solutions are not state of the art. The visualization does not lead to new insights. The major problem of the visualization method is that it produces only blurry images. It may have useful applications in other medical image analysis tasks, where the user has no idea which image features are relevant, for example in prediction of mortality or outcome for some disease and a regular saliency map may not reveal insights.


**Justification Of The Preliminary Rating:**

I would like to argue for accepting this paper because of the strengths identified above.
----------------------------------------------------------------------------------------------------------------------

**Paper Type:**

methodological development

**Special Issue:**

no

---

> ### Author Response · Authors · 2021-03-17
> **Author Response**
>
> Thanks for your feedback! We hope that we have addressed your questions and concerns. We have used your feedback to improve the presentation of the paper!
>
> > A major limitation that is unrelated to the choice of application is that it seems the method cannot generate high-resolution images.
>
> We agree that the autoencoder we used was not very high in resolution, however, our approach will also work with a higher resolution autoencoder or latent variable models.
> In this work, the task we focused on was explaining predictions (by exaggerating or curtailing the features used) by creating the most simple approach possible. With that task in mind, the resolution of our autoencoder was sufficient, especially for the abnormalities that were being evaluated in our study. But a higher resolution model would likely be necessary for explaining small image features used.
>
> > Another more fundamental limitation is that the problem of generating an exaggerated abnormal image is probably ill-posed in most applications: there will usually be many possible ways one can change an image such that it is more likely to be positive for the presence of some abnormality.
>
> We agree! We used the phrase "exaggeration" only to be inline with some related literature. We find that curtailing the features to decrease the prediction yields subjectively looks better and is more in line with the idea of counterfactual generation. We rephrased our task to "counterfactual generation" to make it more clear and changed the title.
>
> > Why did the authors choose to work with the labels atelectasis, cardiomegaly, effusion, and mass?
>
> We chose these based on an intersection between clinical relevance and data available. These abnormalities had models that had been trained on them and there were also expert-drawn bounding-boxes available for these labels.
>
>
> > Now an explanation movie wanting to remove cardiomegaly can do two things: make the heart more slender or the thorax dimensions wider. What is the point of having a movie doing one of the two (the former, on the Github page)? In my opinion, we do not learn anything of practical value.
>
> The value of the movies is so we can exactly have this discussion. The movie is an expression of what the model has learned and not always (although we hope) a true representation of the pathology. It could be that latent shift vector is taking the path of least resistance and changing the heart size is much easier than changing the entire thorax size. We believe that correlating these visualizations with what is expected clinically provides a tool for model introspection and an ability to better understand the (likely) spurious correlations at play.
> A key thing we learned is that the Pneumothorax prediction from the model we evaluated is likely wrong although despite a moderate AUC, which is suggestive of spurious correlations.

---

### Meta-Review · Area_Chair1 · 2021-03-25

**Recommendation:** Accept (Oral & Special Issue & Best P…

**Metareview:**

really strong paper and also super high quality reviewers/reviews. also high-quality discussions during rebuttal.

**Paper Type:**

methodological development

---

### Decision · Program_Chairs · 2021-03-31

**Decision:**

Accept

**Comment:**

Congratulations your paper has been selected as a long oral.